# Temperamental Change in Adolescence and Its Predictive Role on Anxious Symptomatology

**DOI:** 10.3390/bs12060194

**Published:** 2022-06-16

**Authors:** Maria Balle, Aina Fiol-Veny, Alejandro de la Torre-Luque, Jordi Llabres, Xavier Bornas

**Affiliations:** 1University Research Institute on Health Sciences (IUNICS), University of the Balearic Islands, 07122 Palma, Spain; ballemaria@gmail.com (M.B.); ainafiol126@gmail.com (A.F.-V.); jordi.llabres@uib.cat (J.L.); xavier.bornas@gmail.com (X.B.); 2Department of Legal Medicine, Psychiatry and Pathology, CIBERSAM, Complutense University of Madrid, 28040 Madrid, Spain

**Keywords:** temperamental change, adolescence, anxiety, vulnerability

## Abstract

Growing evidence supports the hypothesis that temperamental traits are not static throughout adolescence. The known links between both reactive and regulatory temperament and anxiety symptoms should be investigated bearing this hypothesis in mind. This study collected self-reported data on behavioral inhibition system (BIS) sensitivity, attentional control (AC), and anxiety symptomatology, from 296 adolescents (64.2% girls; M = 12.96 years at the first assessment, SD = 0.47) every six months, four times over eighteen months. The relationships between temperament factors (AC and BIS sensitivity), considered longitudinally (by means of their trajectories) and anxiety symptoms were investigated using Multigroup Latent Growth Modeling (MLGM), as well as the mediating effect of sex on trajectories and anxiety. BIS sensitivity decreased over time and showed differential patterns across sexes. AC remained relatively stable and we found no sex influence on its trajectory. On the other hand, we observed that the BIS sensitivity trajectory was a significant predictor of anxiety symptomatology at age 15. In conclusion, temperamental changes between the ages of 13 and 15 seem to play a relevant role in explaining subsequent anxiety symptomatology, under the mediating influence of sex.

## 1. Introduction

The interaction between reactive and regulatory temperamental processes appears to be essential for a better understanding of internalizing problems [1,2,3]. Reactivity processes are involved in the basic dimensions of emotionality, activity, and sociability, and have been described in terms of valence (e.g., positive emotionality, negative emotionality) as well as motivation (e.g., behavioral activation, behavioral inhibition) [4]. Negative emotionality and behavioral inhibition have been linked to the neuropsychological Behavioral Inhibition System (BIS), whereas positive emotionality and behavioral activation would be related to the Behavioral Activation System (BAS), both core elements of Gray’s reinforcement sensitivity theory of personality [5,6,7]. The BIS is responsible for the detection and resolution of goal–conflict between the BAS (responsive to positive stimuli) and the fight–flight–freeze system (FFFS, a third element of Gray’s revised theory) [8], by inhibiting behavior, increasing arousal, and assessing for risk. According to Cummings, Caporino, and Kendall [9], BIS sensitivity may be a common risk factor for anxiety and depression, whereas BAS hypoactivity may be specific to depression with anhedonic symptoms in children and adolescents. In fact, high BIS sensitivity has been associated with both anxiety and depression problems in youth [10,11,12]. In turn, FFFS sensitivity has been related to fear [13,14].

Regulatory aspects of temperament refer to effortful control and allow for the flexible regulation of reactivity [2]. Attentional Control (AC) constitutes a specific component of the overarching trait of effortful control and refers to the cognitive ability to regulate attentional allocation [15]. Deficits in AC seem to be involved in the etiology of anxiety disorders, since, for example, individuals low in AC are less able to direct attention away from anxiety-provoking stimuli [16,17,18,19].

Most current theories of human development assume some stability in temperamental traits throughout childhood and adolescence. However, the idea that traits are not developmentally static has gained ground over the last few decades [20,21,22]. Adolescence seems to be a period of major changes in those traits (see Caspi, Roberts, & Shiner, 2005 [23], for a review). Existing evidence points to a decreasing pattern in fear, frustration, affiliation, and shyness, and an increasing pattern in high-intensity pleasure in adolescents from 11 to 16 years old with regard to reactive temperament [24,25]. In terms of changes in BIS sensitivity, Urosevic et al. [26] found an increasing pattern from early adolescence to young adulthood, only in girls. Similarly, a cross-sectional study from Pagliaccio et al. [27] identified normative differences in BIS sensitivity scores across age and by sex: scores increased with age throughout early development until age 20–25 years; sex differences emerged in late adolescence, with higher scores in females. The authors discussed their findings within the framework of developmental psychopathology and suggested the need to investigate normative gender differences emerging in BIS sensitivity throughout development as a potential risk for psychopathologies. Finally, the mediation hypothesis of sex was posited, suggesting that sex differences in anxiety may be explained by the fact that several risk factors of anxiety (for instance, temperamental risk factors, among others) are more prevalent in females than in males [28]. In other words, anxiety risk factors would be influenced by sex-related and/or gender-related processes, which, in turn, may lead to a relevant impact on anxiety symptoms.

Less research has considered the potential developmental changes in regulatory temperament during adolescence. Vijayakumar et al. [29] found that effortful control remained stable between early and mid-adolescence in males but decreased over time in females. Meanwhile, Laceulle et al. [24]. observed this decreasing pattern in effortful control in both boys and girls from ages 11 to 16 years. Atherton et al. [30] did not find either differences based on sex, but in their case, they observed a different pattern in effortful control: a slight decline from 10 to 14 years and a rapid increase from age 16 to 19. Besides, these authors focused on the age-related changes of effortful control components and noted that AC decreased linearly from age 10 to 19. Although developmental study of effortful control on adolescence has begun to emerge, to our knowledge, the study of Atherton and their colleagues is the only one that has focused on the AC trajectory during adolescence.

The studies reviewed so far employed long (usually 12-month) timescales to cover long periods of development (including entire adolescence and sometimes even further) and, therefore, changes occurring at shorter timescales could go unnoticed. Increases in BIS sensitivity or AC that take place over a timescale of a couple of months, for instance, can be masked by a general decreasing trend across five or six years of development. There is no information on temperament changes at timescales under one year to formulate specific hypotheses as to the expected changes during these periods; however, just as increasing the resolution of an image enables us to see more details, using shorter timescales would allow us to appreciate changes beyond the general trend. Further, investigating changes at these short timescales is as relevant for diagnosing and preventing the onset of anxiety disorders as research that uses longer timescales. Indeed, anxiety disorders may appear at any time during adolescence.

Studies examining the contributions of both reactive and regulatory temperament factors to youth anxiety have multiplied in recent decades (e.g., [18,19,31,32,33]), but research into the specific contribution of BIS sensitivity and AC has been scarce. The combination of high BIS sensitivity and low AC has been associated with high levels of anxiety symptomatology in non-clinical adolescents [17], but a longitudinal approach has scarcely been followed. Sportel et al. [34] focused on BIS, FFFS, and AC as potential predictors of internalizing symptoms over a two-year period (participants were aged between 12 and 15) and concluded that high BIS, high FFFS, and low AC had a cumulative predictive role for future anxiety symptoms in adolescents, although this effect disappeared after baseline anxiety levels were controlled.

Within the developmental psychopathology framework, Laceulle et al. [35] investigated whether changes in adolescent temperament that reflect the opposite of the maturational normative (see [24]) may put individuals at risk of mental disorders. Analyses revealed that an increasing pattern in fear and frustration emotionality between ages 11 and 16 predicted internalizing disorders between ages 16 and 19. Moreover, although basal levels of effortful control predicted internalizing disorders, changes in this variable were not related with affective disorders. Despite the well-established gender differences, regarding both temperament and mental health problems, Laceulle et al. [35] found no differences between boys and girls in temperament change–psychopathological diagnoses associations.

Given mixed findings on reactive and regulatory temperament trajectories and sex differences, we think that additional work is needed to better understand their role on anxious symptomatology. The first aim of this study was to examine the developmental trajectories of reactive (BIS sensitivity) and regulatory (AC) temperament among 13 and 15 year old participants, using a 6-month timescale (i.e., through four assessments, one every six months). Given the inconsistent temperamental gender differences in the literature, we studied the role of sex on developmental trajectories. The second aim of this study was to test whether temperamental trajectories would predict anxiety symptomatology at age 15, considering sex differences. We hypothesized that increasing trajectories or persistently high scores of BIS sensitivity and decreasing trajectories or persistently low scores of AC would be associated with high overall anxiety at age 15, for both boys and girls.

## 2. Materials and Methods

### 2.1. Participants

The initial sample was composed of 752 early adolescent participants (56% female) from 20 high schools in the same school district who agreed to participate in the study. The mean age of the adolescents at the start of the study was 13.0 years (SD = 0.56). The youths were all Caucasian, had middle socioeconomic statuses, and lived in both urban and rural areas. The participants and their parents/tutors provided written consent.

The project “Complex trajectories of anxiety in adolescence: towards a better prediction of the onset of anxiety disorders” was approved by the University Bioethics Committee of the University of the Balearic Islands.

### 2.2. Measures

#### 2.2.1. Anxiety Symptomatology

The Revised Child Anxiety and Depression Scale (RCADS) [36,37] was used to assess anxiety symptoms. It is a 47-item self-report questionnaire, with scales corresponding to separation anxiety disorder; social phobia; generalized anxiety disorder; panic disorder; obsessive compulsive disorder; and major depressive disorder. It provides a scale indicating the total level of anxiety symptomatology. For the purpose of the current study, only this total scale was used. The RCADS requires respondents to rate how often each item applies to them. Items are scored 0–3 corresponding to “never”, “sometimes”, “often”, and “always”. The internal consistency was α = 0.94 for the total anxiety scale.

#### 2.2.2. BIS Sensitivity

The Sensitivity to Punishment scale of the SPSRQ-J [38] was used as a self-report measure of BIS sensitivity. This is a self-report instrument adapted for children and adolescents from the adult version developed by Torrubia et al. [39] The SPSRQ-J is a 30-item yes–no questionnaire containing two scales of 15 items each: Sensitivity to Punishment and Sensitivity to Reward. Scores for each scale were obtained by adding up all “yes” answers. In the current study, only the Sensitivity to Punishment scale was relevant to the hypothesis and, thus, used in the analyses. This scale was “designed to measure individual differences in some functions dependent on the BIS in checking and controlling modes: (1) behavioral inhibition (passive avoidance) in general situations involving the possibility of aversive consequences or novelty, and (2) worry or cognitive processes produced by threat of punishment or failure” ([39], p. 844). That is, the questionnaire retains the assessment of cognitive anxiety responses when one is facing threatening conditions (from which the use of the term “sensitivity to punishment” derives), adding items asking about passive avoidance tendencies. Internal consistency was found to be good across all assessment times, with Cronbach αs between 0.76 and 0.82. Between-point measurement correlations ranged from r = 0.56 to r = 0.76.

#### 2.2.3. Attentional Control

The Early Adolescence Temperament Questionnaire Revised (EATQ-R) [40,41] is a 103-item self-report measure of 12 different domains related to adolescent temperament that comprises four principal factors: Negative Affectivity, Effortful Control, Affiliation, and Surgency. Each item has to be answered on a 5-point Likert scale ranging from 1 = almost never true to 5 = almost always true. Total trait and factor scores can be computed by adding up ratings across relevant items (after recoding inversely formulated items). The data presented here focus on the Attentional Control subscale (a first-order factor saturating on the Effortful Control principal factor), with Cronbach αs between 0.50 and 0.56, and correlations between time assessments from r = 0.48 to r = 0.64.

### 2.3. Procedure

Twenty-three high schools in the school district were invited to participate in the study. Twenty schools agreed to do so. Before the start of the study, participants and their parents received written information and attended a meeting where they were informed about the study. The assessments began after participating families had provided written informed consent. Measures were administered through paper-and-pencil protocol. For BIS sensitivity and AC, missing data for each measurement point were different: a sample of 520 participants (69.14% of our initial sample) filled out both questionnaires at Time 1 (T1); 646 participants at Time 2 (T2), 649 at Time 3 (T3), and 515 participants at Time 4 (T4). Anxiety symptomatology was assessed at T4. A total of 296 participants (64.2% girls; M = 12.96 years old at T1, SD = 0.47) filled out all the questionnaires at all measurement points.

### 2.4. Statistical Analyses

Multigroup Latent Growth Modeling (MLGM) [42,43] was used to test the influence of the temperamental factor trajectories (i.e., BIS sensitivity and AC) on anxiety symptomatology at T4, and the mediating effect of sex. MLGMs capture individual differences in developmental trajectories by including variances for latent factors, considering varying non-overlapping and mutually exclusive groups (boys and girls) with potentially different score distributions. According to latent growth curve tradition, developmental trajectory of each temperamental factor was represented by two latent factors: mean initial level (intercept) and mean change over time (slope). First, we explored whether the temperamental factors showed either a linear growth or a quadratic growth across our study period. Model parameters were estimated using the robust maximum likelihood algorithm.

To satisfy the first study aim, we tested the fit of a latent factor model covering the developmental trajectories of AC and BIS on anxiety, and assuming no sex effect either on temperament trajectories or anxiety (unconstrained model). To satisfy the second study aim, we compared nested models considering the effect of sex on different paths (i.e., BIS sensitivity path and AC path), all of them covering the effect of the temperament trajectories on anxiety. Sex effect was tested by constraining model parameters across groups. Thus, a model considering the mediating effect of sex just on temperament trajectories (configural model) was studied following the measurement invariance tradition. Secondly, constraints on mean and error variance of latent factors were imposed (parameter constraints on observable factors were discarded due to the nature of MLGM models), on AC trajectory parameters, BIS, or both factors. When the constraints were imposed to only one path (either AC trajectory parameters or BIS) partial measurement invariance was tested. Finally, the fit of a model comprising the regressing effect of the latent factors on anxiety under the mediating effect of sex (full model) was studied.

Five indices were selected to test the fit of each individual model: the scaled χ^2^ statistic for goodness of fit [44]; the robust root mean square error of approximation index (RMSEA); the robust comparative fit index (CFI); the robust Tucker–Lewis index (TLI); and the standardized root mean square residual (SRMR). Reasonable approximate fit is proven when RMSEA < 0.080; CFI ≥ 0.95, TLI ≥ 0.95, and SRMR < 0.080. Additionally, we conducted comparisons between the unconstrained model and the nested models by means of the incremental RMSEA (ΔRMSEA), incremental SRMR (ΔSRMR), and incremental CFI (ΔCFI). An ΔCFI ≥ −0.010 and ΔRMSEA ≥ 0.015 or ΔSRMR ≥ 0.010 would reflect statistically significant differences between nested models [45,46], since the simpler one would not be nested within the more complex one (and, hence, a sex-related influence was endorsed).

Analyses were run using the IBM SPSS Statistics v. 25.0.0 statistical package and R x64 v. 3.3.0 (package lavaan 0.5–20).

## 3. Results

Figure 1 displays the BIS sensitivity and AC trajectories for boys and girls from T1 to T4.

In general terms, BIS sensitivity showed a statistically significant effect, F (3, 803.74) = 14.82, *p* < 0.001, η^2^_partial_ = 0.048, observing a decreasing trend in girls from T1 to T4, t (295) = 6.10, *p* < 0.001, Hedges’ g = 0.43; and boys from T2 to T4, t (295) = 2.96, *p* = 0.020, Hedges’ g = 0.29. Despite developmental changes, girls scored higher than boys in BIS sensitivity across the measurement moments, except at T3. For AC, there were no differences between sexes. In terms of anxiety level, girls showed higher levels in the total anxiety score (for boys, M = 23.50, SD = 13.95; and for girls, M = 27.63, SD = 16.73), with t (251.19) = −2.27, *p* = 0.024, Hedges’ g = −0.26.

Regarding the growth curve model, the solution depicting linear growth in the temperamental factors fitted better to the data (χ^2^ (26) = 54.79, *p* < 0.001; CFI = 0.98, TLI = 0.97, RMSEA = 0.062, SRMR = 0.039), in comparison to the model depicting a quadratic growth (χ^2^ (26) = 60.55, *p* < 0.001; CFI = 0.97, TLI = 0.96, RMSEA = 0.067, SRMR = 0.049). Only the BIS latent factors showed a statistically significant regressing effect on the anxiety score, both factors, the latent intercept, B = 3.30, SE = 0.30, z = 11.03, *p* < 0.001, and the latent slope, B = 7.72, SE = 2.01, z = 3.85, *p* < 0.001. These results indicate that the higher the BIS scores across time points, the higher the anxiety symptoms; the anxiety symptoms may be greater with a more increasing BIS course.

The results from MLGM analysis showed the lack of measurement invariance at the mean and mean + variance level (see Table 1). Parameters of the configural model by sex are displayed in the Appendix A. The lack of the measurement invariance (i.e., pointing to sex-related differences) was evident at the latent mean scores only for the BIS path (CFI = −0.008 and ΔSRMR = 0.017). More concretely, girls showed a statistically significant latent intercept, B = 4.53, SE = 0.24, z = 30.12, *p* < 0.001 and a statistically significant negative latent slope, B = −0.49, SE = 0.08, z = −5.81, *p* < 0.001. Conversely, boys showed a higher latent intercept, B = 5.88, SE = 0.30, z = 19.38, *p* < 0.001 and a negative latent slope, B = −0.26, SE = 0.11, z = −2.29, *p* = 0.022, with a lower decreasing trend. The latent variance models showed a better fit to the simpler nested model when constraints were imposed to parameters from both AC and BIS paths (CFI = 0.002 and ΔSRMR = 0.019).

Sex-specific predicted values on both latent intercepts (i.e., AC and BIS) were used to identify participants at higher temperamental risk of elevated anxiety symptoms (i.e., those with BIS latent intercept over the 75th distribution percentile and AC latent intercept below the 25th distribution percentile). As a result, a total of 36 participants (12.16% of sample; *N* = 25 female) was classified into the group of combined risk. These participants showed significantly higher anxiety symptoms (M = 37.92, SD = 18.87) than participants without risk (i.e., those with BIS latent intercept below the 25th distribution percentile and AC latent intercept above the 75th distribution percentile; M = 21.71, SD = 13.83), t (217) = 6.02, *p* < 0.001, Hedges’ g = 1.10.

## 4. Discussion

The aim of this study was twofold: to examine how the developmental trajectories of reactive and regulatory temperament between the ages of 13 and 15 years could predict anxiety symptomatology at age 15 and to explore whether sex showed mediating effects on anxiety, directly or indirectly (through its influence on the temperament trajectories).

In terms of the first aim, the six-month timeframe constitutes an optimal interval to gain accuracy in depicting the course of temperamental traits throughout our assessment period in adolescence. At higher time resolution, changes are not so smooth. The overall trend of BIS sensitivity throughout the four assessments was decreasing in girls from T1 and T4, remaining stable from T1 to T3 to later drop in boys. Furthermore, BIS sensitivity scores for girls were higher than the scores from boys at every assessment time point.

Moreover, we found that only the course of the BIS influenced on the development of anxiety symptoms. This may be related to the overall influence of BIS sensitivity to anxiety symptoms, regardless of types [6,11]. In turn, a mediating effect of sex on the developmental trajectory of BIS sensitivity was observed. In this regard, we found a lack of measurement invariance when considering rank distributions across both sexes. Sex influence on BIS sensitivity has already been supported in prior findings. Thus, girls usually exhibit higher BIS sensitivity compared to boys [17,19,26,27]. In the same vein, and at a more general level, females report greater-negative affectivity and neuroticism [47,48]. Several mechanisms may explain these gender differences. From a psychobiological perspective, Li et al. [49] suggested that BIS sensitivity differences might be partially due to sex-specific neuroanatomical differences in the brain structures involved in the processing of negative emotions (i.e., parahippocampal gyrus). Attenuated cardiac complexity, as a sign of worse adjustment to contextual demands, was associated with higher risk of anxiety in girls [50]. On the other hand, Brody’s theory of gender differences in emotion supports that early gender differences in child temperament (i.e., BIS sensitivity) elicit parents’ differential socialization of girls and boys, which, in turn, magnifies preexisting sex differences in trait emotion [51].

The above-mentioned decreasing pattern has not been consistently observed in previous studies. In their longitudinal study, Urosevic et al. [26] (2012, Appendix A) assessed BIS sensitivity in individuals ranging in age from 9 to 23 years, across a two-year follow-up. Although the overall pattern from 9 to 23 years old was increasing, in the late adolescents’ group (i.e., ages 13–17 at Time 1 and ages 15–19 at Time 2), corresponding to our sample age range, no differences were observed between T1 and T2, suggesting a consistency of this variable at these ages, in line with previous studies from our research team [19]. Using a laboratory task, Colder et al. [52] examined trajectories of sensitivity to reward and sensitivity to punishment (i.e., BIS sensitivity) across three annual assessments as predictors of escalation of early substance use. The mean baseline age of adolescents was 11.8 years. In terms of changes in sensitivity to punishment, the authors observed that this variable declined on average over time, although there were no individual differences in rates of change. As noted earlier, the use of such a long timescale (two years or twelve months) might have masked subtle changes that take place over shorter periods.

AC remained relatively stable over time and its course showed a lack of influence on anxiety symptom development. This may be explained in light of the results of other existing studies; the AC may be a specific risk factor of the development of particular types of anxiety symptoms, such as social anxiety or generalized anxiety [53,54]. On the other hand, no differences between boys and girls were found, except at Time 3 where girls scored higher than boys. The lack of differences was confirmed by the latent curve models. The consistency of AC may be explained as a part of effortful control. Laceulle et al. [24] did not find changes in effortful control in the expected direction and suggested that “It might be that effortful control matures late compared with other traits or that it even shows a temporary relapse effect between ages 11 and 16 years” (p. 281). Sex did not show a mediating effect on the trajectory of AC in terms of the latent factors depicting the developmental course, but it did in terms of how measurement points covariate across measurement points. In this regard, the mean level of AC across assessment points (the intercept latent construct of the AC trajectory) explained with a negative loading the anxiety symptomatology at T4, similarly in girls and boys.

Our findings should be interpreted with some caution due to limitations. First, some extrinsic factors that may influence anxiety were not addressed in this study (e.g., experiencing a traumatic event, daily stress burden, family interactions, etc.). This study focused on temperament vulnerability with the aim of delimitating its actual, non-confounded influence on anxiety. Future research should cover a wide variety of factors, extrinsic and intrinsic, in order to provide a more complete picture of vulnerability and risk factors on adolescent anxiety. Second, some objective concomitants of temperament could be incorporated (e.g., cardiovascular function, hormonal release, etc.) in order to gather more exhaustive evidence on the influence of vulnerability factors in anxiety. Further studies should provide this wider perspective to study the effects of vulnerability on anxiety. On the other hand, we could only analyze data from 296 participants out of our initial sample (*N* = 752). This was because we only used data from complete cases (i.e., participants without any missing assessment point), taking into account the difficulties associated with a longitudinal, repeated-measure design. Further details on response rate across measurement points can be consulted elsewhere [19]. On the other hand, the sample size was enough to detect a moderate effect size (δ = 0.30) on explaining the outcome (i.e., anxiety symptoms), with power, 1 − β = 0.80, α = 0.05, four latent variables (i.e., latent intercepts and slopes from the temperamental traits), and six observed variables. The minimum sample size needed to see the desired effect size is 137, according to an a priori sample size calculation conducted using the Soper’s software [55]. Finally, we used a subscale with relatively low reliability (the EATQ-R Attentional Control subscale). However, we were particularly interested in assessing attentional control specifically (but not the higher-order construct, the Effortful Control), taking into account the results by previous studies [16,17]. The low reliability of the scale may compromise the results from this study. Unfortunately, other scales on attentional control for adolescents show low reliability levels [17,56]. Further studies should be conducted using other tools (e.g., experimental paradigms on cognitive performance) to provide more accurate findings on the role of attentional control.

## 5. Conclusions

We think these findings are of great interest for several reasons. First, our study is among the few to investigate the consequences of temperament change on psychopathology. In this regard, we observed that changes in BIS sensitivity over time predict anxiety symptomatology. Thus, higher BIS sensitivity scores and low decreases (as we found a decreasing BIS sensitivity trajectory) across assessment points predicted higher anxiety scores at T4. As a practical implication derived from these results, we recommend fluctuations on temperamental traits to be monitored throughout adolescence to identify venues for mental health condition onset. To date, research has been conducted on the effects of the interaction between different temperamental characteristics and youth anxiety symptoms, but to our knowledge, the usefulness of the interplay of reactive and regulatory temperamental characteristics and sex influence on it have yet to be studied. On the other hand, our study aims at monitoring temperamental change across two years, in a critical period of life for the development of personality, with four measuring points that provide a more accurate picture about the developmental course of psychopathology. Semestral assessment, therefore, seems to be an optimal timeframe to visualize temperamental change in adolescence and its subsequent effect on anxiety symptom course.

## Figures and Tables

**Figure 1 behavsci-12-00194-f001:**
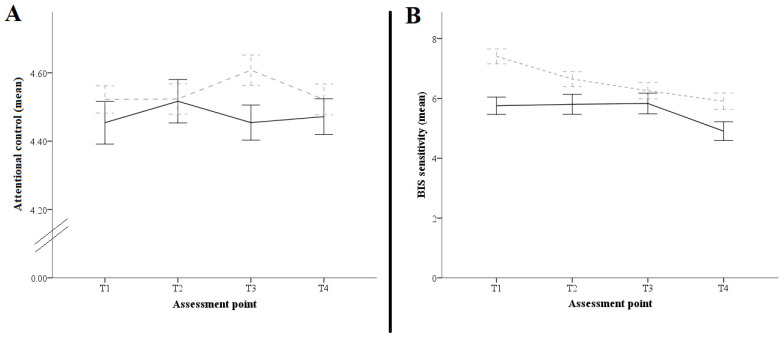
Temperament trajectories according to sex. Box (**A**) shows the attention control trajectories over our assessment period. Box (**B**) displays the behavioral inhibition system sensitivity trajectories over our assessment period. Solid line depicts the trajectories of boys. Dashed line depicts the trajectories of girls. BIS = Behavioral inhibition system. Error bars depict the standard error of the mean.

**Table 1 behavsci-12-00194-t001:** Measurement invariance model fit summary for RCADS Total anxiety.

	Scaled χ^2^ (df)	RMSEA	CFI	TLI	SRMR	Model Comparison		
Model						ΔCFI	ΔRMSEA	ΔSRMR
Configural	**109.22 (52)**	**0.086**	**0.96**	**0.94**	**0.056**			
Latent mean MI	127.75 (56)	0.093	0.95	0.93	0.073	−0.011	0.007	0.017
Partial mean MI (BIS)	**121.99 (54)**	**0.092**	**0.95**	**0.93**	**0.073**	**−0.008**	**0.006**	**0.017**
Partial mean MI (AC)	110.92 (54)	0.084	0.96	0.94	0.058	0.001	−0.002	0.002
Latent variance MI	**129.16 (60)**	**0.088**	**0.95**	**0.94**	**0.075**	**0.002**	**−0.005**	**0.019**
Partial variance MI (BIS)	122.80 (56)	0.090	0.95	0.93	0.075	0.001	−0.004	0.002
Partial variance MI (AC)	111.66 (56)	0.082	0.96	0.95	0.060	−0.010	−0.010	−0.013
Full MI	137.87 (64)	0.087	0.95	0.94	0.077	−0.001	−0.001	0.002
Partial full MI (BIS)	129.05 (58)	0.089	0.95	0.94	0.076	0.001	0.001	0.001
Partial full MI (AC)	112.89 (58)	0.079	0.96	0.95	0.061	0.012	−0.009	−0.014

Note. All χ^2^-based tests were statistically significant with a *p* < 0.01. Selected models are diplayed in bold face. Each nested model was compared with the simpler one selected. The measurement invariance models were increasingly constrained: latent mean, latent variance (latent means + variances), full MI (latent means + variances + regressors). The partial MI models involved constraints on either BIS path parameters or AC path parameters. MI = Measurement invariance. BIS = Behavioral inhibition system. AC = Attentional control. df = degrees of freedom. RMSEA = root mean square error of approximation index. CFI = comparative fit index. TLI = Tucker-Lewis index. SRMR = standardised root mean square residual. ΔCFI = incremental CFI. ΔRMSEA = incremental RMSEA. ΔSRMR = incremental SRMR.

## Data Availability

Data are not publicly available (they could be accessed under reasoned request). Analyses may be provided under request to corresponding author.

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
