# Peer review of "Temperamental Change in Adolescence and Its Predictive Role on Anxious Symptomatology"

_behavsci, 2022, doi:10.3390/bs12060194_

Round 1

Reviewer 1 Report

Dear authors,

Firstly, I would like to thank you for the opportunity to review your paper. It focuses on a quite interesting topic and is supported by a strong rationale. The statistical analysis was conducted with rigor. Thus, I would like to congratulate you for your research and for this manuscript. 

Nonetheless, there are some few issues that I think might be addressed:

a) It is not clear if all the assessment tools you used were validated for the Spanish population (namely the RCADS and EATQ-R). If they are not, that should be reported as a significant limitation of your study.

b) You started your study with a large sample, but only 296 participants answered the questionnaires at all measurement points. There is, clearly, a low retention rate. Don't you think that may also be a limitation of your study?

c) Moreover, you did not present any sample size calculation. Is 296 a good sample size to answer the objectives of your study? Is this sample size "enough" to generalize the findings? If not, that is also an important limitation of your study.

d) Finally, you should avoid references in the "Conclusions" section. On the other side, you should clearly state the implications of your study and present some recommendations for further research on this domain.

Author Response

Dear reviewer,

We have just submitted the revised version of our manuscript “Temperamental Change in Adolescence and its Predictive Role on Anxious Symptomatology” by Dr Balle et al. for publication as a research paper in the leading journal, Behavioral Sciences. The authors would like to thank for the time you devoted to review our manuscript.

This revised manuscript includes changes in line with your comments and suggestions. Please, see below details on how we addressed your comments.

Kind regards.

Alejandro de la Torre-Luque, PhD

Department of Legal Medicine, Psychiatry and Patology.

Universidad Complutense de Madrid

Madrid, Spain

RESPONSE TO REVIEWER'S COMMENTS:

a) It is not clear if all the assessment tools you used were validated for the Spanish population (namely the RCADS and EATQ-R). If they are not, that should be reported as a significant limitation of your study.

- Thank you for this comment. We included the reference to the validation study of both RCADS (ref. 36) and EATQ-R (ref. 40) Spanish versions. Please, note that the validation of the SPSRQ-J’s Spanish version had been provided in the main manuscript submitted (ref. 38).

b) You started your study with a large sample, but only 296 participants answered the questionnaires at all measurement points. There is, clearly, a low retention rate. Don't you think that may also be a limitation of your study?

- We mentioned the low retention letter as a limitation of this study (please, note that the sample size was enough to conduct the analysis: see my response to your next comment). Please, see from line 343 onwards.

c) Moreover, you did not present any sample size calculation. Is 296 a good sample size to answer the objectives of your study? Is this sample size “enough” to generalize the findings? If not, that is also an important limitation of your study.

- In line with the previous comment, we included the a priori sample size conducted to support that the sample size is enough to visualize the desired effect size for the main outcome (i.e., anxiety symptoms). Please, see from line 347 onwards.

d) Finally, you should avoid references in the "Conclusions" section. On the other side, you should clearly state the implications of your study and present some recommendations for further research on this domain.

- The Conclusions section was rewrote to make clearer the take-home message from the study as well as to extend the practical implications derived from the study. Finally, references were removed from this section. Please, see from line 361 onwards.

Reviewer 2 Report

I was pleased to get the opportunity to review your paper. The topic is very important, temperament in children and adolescents can predict behavioural abnormalities. Interventions targeting the regulatory components of temperament may reduce the development of different mental disorders symptoms. A methodologically and statistically strong manuscript. I have suggested to the authors that give some practical significance to their results.

There are some minor recommendations to improve the manuscript.

- It is recommended to add a List of Abbreviations.

- Line 133. RCADS, this is the first mention, please add what RCADS means.

-  Line 159-166. Please use the same name for the subscales. Is the Effortful control the same as the Attentional Control subscale? Cronbach alpha for this subscale is very low, almost unacceptable. How do you explain this result? How is the reliability of the analyses affected?

- Statistical analysis. Based on the literature, I'm not sure that the mediating effect of gender was a good assumption. Why the mediating effect was examined instead of the moderating effect. Why examine both linear and quadratic growth? What is the reason why a quadratic growth can be assumed? Please give some more explanation of these, especially for those who are not so familiar with statistical modelling.

- Results. Line 223. …” girls always scored higher than boys in BIS sensitivity. In Figure 1 at T3, I think, that the score of the girls and boys did not differ. In Figure 1, please correct Attention control to Attentional Control. Please do not distort the y-axis. The two figures (A and B) next to each other are misleading. Please add the exact p-value, and it is recommended to use an unbiased effect size such as Hedges’ g (there were more girls than boys in the sample). It is recommended to use ‘statistically significant’ higher…. In the terms of anxiety, I do not think that this difference has any practical significance. How much of a difference can be considered practically significant in this case?

- Discussion. Line 262-263. Please correct this sentence (see comments in Results).

Questions:

-          In the Introduction, the authors mentioned that the combination of high BIS sensitivity and low AC has been associated with high levels of anxiety symptomatology in adolescence. Did you examine this combination? How prevalent is this combination in this sample?

-          It was not clear enough for me where I can find the testing invariance of the growth function.

-          Did you test the invariance of latent growth factor means, i.e., the mean of the initial outcome level at the baseline? In Figure 1 at T1, there is a significant difference between girls and boys.

-          Can your results provide any practical advice for mental health professionals? It would be good to see whether the results of statistical analyses can be put to practical use at this stage of your studies. While it is true that the situation can be much more complex and there are more confounding factors need to be considered, but I was very much missing some practical conclusions, even if only for a few sentences.

Author Response

Dear reviewer,

We have just submitted the revised version of our manuscript “Temperamental Change in Adolescence and its Predictive Role on Anxious Symptomatology” by Dr Balle et al. for publication as a research paper in the leading journal, Behavioral Sciences. The authors would like to thank for the time you devoted to review our manuscript.

This revised manuscript includes changes in line with your comments and suggestions. Please, see below details on how we addressed your comments.

Kind regards.

Alejandro de la Torre-Luque, PhD

Department of Legal Medicine, Psychiatry and Pathology.

Universidad Complutense de Madrid

Madrid, Spain

RESPONSE TO REVIEWER'S COMMENTS:

- It is recommended to add a List of Abbreviations.

** The journal does not provide any guideline regarding the list of abbreviations. For that reason, we decided not including such a list.

- Line 133. RCADS, this is the first mention, please add what RCADS means.

** Done (see in line 139)

-  Line 159-166. Please use the same name for the subscales. Is the Effortful control the same as the Attentional Control subscale? Cronbach alpha for this subscale is very low, almost unacceptable. How do you explain this result? How is the reliability of the analyses affected?

** Thank you for the insightful comment. First, we clarified that the Attentional control (AC) subscale was our focus of analysis. AC constitutes a component of the effortful control temperament dimension. We stated that from line 164-173. On the other hand, the authors agreed that the Cronbach’s alpha is quite low for the AC subscale. We mentioned that in the limitations section (as well as its potential influence on the study results). Please, see the changes from line 352. Unfortunately, other scales on AC for adolescents show low reliability levels. Further studies should be conducted using other tools (e.g., experimental paradigms on cognitive performance) to provide more accurate findings on the role of attentional control.

- Statistical analysis. Based on the literature, I'm not sure that the mediating effect of gender was a good assumption. Why the mediating effect was examined instead of the moderating effect. Why examine both linear and quadratic growth? What is the reason why a quadratic growth can be assumed? Please give some more explanation of these, especially for those who are not so familiar with statistical modelling.

** First, our study comes from the mediation hypothesis of sex influence on anxiety, as explained in the Introduction section (see from line 65 onwards). This means that the sex-related differences in anxiety symptoms may be derived from sex-related differences in risk factors (e.g., temperamental factors).

Second, it is very common to test the shape of the course of a developmental process, taking into account the lack of clear, consistent evidence on that. Several studies have supported that varying temperamental traits may evolve either linearly or exponentially [1,2]. As we have four measurement points linear and quadratic effect of time (age) can be tested.

- Results. Line 223. …” girls always scored higher than boys in BIS sensitivity. In Figure 1 at T3, I think, that the score of the girls and boys did not differ. In Figure 1, please correct Attention control to Attentional Control. Please do not distort the y-axis. The two figures (A and B) next to each other are misleading. Please add the exact p-value, and it is recommended to use an unbiased effect size such as Hedges’ g (there were more girls than boys in the sample). It is recommended to use ‘statistically significant’ higher…. In the terms of anxiety, I do not think that this difference has any practical significance. How much of a difference can be considered practically significant in this case?

** We corrected the sentence and specify some additional statistics to make clearer the study results: exact p value when p > .001, Hedges’ g effect size estimate, the y-axis label ‘Attentional control’ for the Figure 1; as well as the descriptor ‘statistically significant’ when applicable. On the other hand, please note that our interest was not to report sex-related differences in terms of the observable factors, but in terms of the latent factors included in the course of both temperamental factors (as you could see from line 251 onwards). On the other hand, we think that presenting the figure with the trajectory of both temperamental traits is highly informative, as the reader could realize the dynamics of both over time. Regarding sex-related differences in terms of anxiety symptoms, differences were statistically significant with a small effect size (see in line 236-237).

- Discussion. Line 262-263. Please correct this sentence (see comments in Results).

** Done (see from line 278-280)

Questions:

- In the Introduction, the authors mentioned that the combination of high BIS sensitivity and low AC has been associated with high levels of anxiety symptomatology in adolescence. Did you examine this combination? How prevalent is this combination in this sample?

** Highly interesting question. In this regard, it deserves to consider that our longitudinal design involves the traditional combination of risk conditions to be reformulated. We calculated sex-specific predicted values of both latent intercepts (i.e., AC and BIS) were used to identify participants at higher temperamental risk of elevated anxiety symptoms (i.e., those with BIS latent intercept over the 75th distribution percentile and AC latent intercept below the 25th distribution percentile). As a result, a total of 36 participants (12.16% of sample; N = 25 female) was classified into the group of combined risk. These participants showed significantly higher anxiety symptoms (M = 37.92, SD = 18.87) than participants without risk (i.e., those with BIS latent intercept below the 25th distribution percentile and AC latent intercept above the 75th distribution percentile; M = 21.71, SD = 13.83), t (217) = 6.02, p < .001, Hedges’ g = 1.10. All these results were included within the Results section (lines 262-269)

- It was not clear enough for me where I can find the testing invariance of the growth function.

- Did you test the invariance of latent growth factor means, i.e., the mean of the initial outcome level at the baseline? In Figure 1 at T1, there is a significant difference between girls and boys.

** Under the latent growth tradition, a longitudinal (developmental) process is defined by two constructs: the latent intercept (mean level of this construct across the measurement occasions) and the latent slope (change dynamics across measurement occasions) [3]. Parameters that connect observed variables with the latent factors (i.e., observed loadings) are fixed in order to preserve model identification in terms of the structural model. On the other hand, to test the mediation hypothesis, parameters at the latent level should be fixed to be equal between sexes: latent means (roughly, means of the latent factors), latent variance (estimation error of latent variables) and regressors (loading of the latent factors on the endogenous factor, or the outcome: anxiety symptoms). Therefore, we tested the mediation effect of sex on the latent constructs of both temperamental trajectories (latent means), the construct variability (estimation error or residual) and on the role of each temperamental trajectory in anxiety symptoms (regressors). On the other hand, we presented the observed scores but not the latent scores in the Figure 1. The latent growth model provides far more details on the longitudinal process that the mere observed measures, as it takes into account the process, by their two key dimensions (mean level and trajectory).

- Can your results provide any practical advice for mental health professionals? It would be good to see whether the results of statistical analyses can be put to practical use at this stage of your studies. While it is true that the situation can be much more complex and there are more confounding factors need to be considered, but I was very much missing some practical conclusions, even if only for a few sentences.

** The Conclusions section was rewrote to make clearer the take-home message from the study as well as to extend the practical implications derived from the study. Finally, references were removed from this section. Please, see from line 361 onwards.

REFERENCE

  1. Mathesius, J.R.; Lussier, P.; Corrado, R.R. The Early Temperamental Correlates of Antisocial Propensity. J. Crim. Justice 2020, 66, 101630, doi:https://doi.org/10.1016/j.jcrimjus.2019.101630.
  2. Atherton, O.E.; Lawson, K.M.; Robins, R.W. The Development of Effortful Control from Late Childhood to Young Adulthood. J. Pers. Soc. Psychol. 2020, 119, 417–456, doi:10.1037/pspp0000283.
  3. Grimm, K.J.; Ram, N. Latent Growth and Dynamic Structural Equation Models. Ssrn 2018, doi:10.1146/annurev-clinpsy-050817-084840.